# Silver Nanostructured Substrates in LDI-MS of Low Molecular Weight Compounds

**DOI:** 10.3390/ma15134660

**Published:** 2022-07-02

**Authors:** Gulyaim Sagandykova, Piotr Piszczek, Aleksandra Radtke, Radik Mametov, Oleksandra Pryshchepa, Dorota Gabryś, Mateusz Kolankowski, Paweł Pomastowski

**Affiliations:** 1Centre for Modern Interdisciplinary Technologies, Nicolaus Copernicus University in Toruń, Wileńska 4, 87-100 Toruń, Poland; mametov.radik@gmail.com (R.M.); pryshchepa.alexie@gmail.com (O.P.); mate.kola@hotmail.com (M.K.); pomastowski.pawel@gmail.com (P.P.); 2Department of Inorganic and Coordination Chemistry, Faculty of Chemistry, Nicolaus Copernicus University in Toruń, Gagarina 7, 87-100 Toruń, Poland; piszczek@umk.pl (P.P.); aradtke@umk.pl (A.R.); 3Department of Environmental Chemistry and Bioanalytics, Faculty of Chemistry, Nicolaus Copernicus University in Toruń, 87-100 Toruń, Poland; 4Radiotherapy Department, Maria Sklodowska-Curie National Research Institute of Oncology, Gliwice Branch, Wybrzeże Armii Krajowej 15, 44-101 Gliwice, Poland; dorota.gabrys@io.gliwice.pl

**Keywords:** laser desorption/ionization mass spectrometry, small biomolecules, silver nanostructures, chemical vapor deposition

## Abstract

Mass spectrometric techniques can provide data on the composition of a studied sample, utilizing both targeted and untargeted approaches to solve various research problems. Analysis of compounds in the low mass range has practical implications in many areas of research and industry. Laser desorption ionization techniques are utilized for the analysis of molecules in a low mass region using low sample volume, providing high sensitivity with low chemical background. The fabrication of substrates based on nanostructures to assist ionization with well-controlled morphology may improve LDI-MS efficiency for silver nanoparticles with plasmonic properties. In this work, we report an approach for the preparation of silver nanostructured substrates applied as laser desorption ionization (LDI) plates, using the chemical vapor deposition (CVD) technique. Depending on the mass of used CVD precursor, the approach allowed the synthesis of LDI plates with tunable sensitivity for various low molecular weight compounds in both ion-positive and ion-negative modes. Reduced chemical background and sensitivity to small biomolecules of various classes (fatty acids, amino acids and water-soluble metabolites) at nanomolar and picomolar detection levels for lipids such as triacylglycerols, phosphatidylethanolamines and lyso-phosphatidylcholines represent an emerging perspective for applications of LDI-MS plates for the collection of molecular profiles and targeted analysis of low molecular weight compounds for various purposes.

## 1. Introduction

Low molecular weight (LMW) compounds are targeted in various research fields with the aims of (i) searching for potential diseases biomarkers, e.g., gastric or prostate cancers [1,2], (ii) safety control of food products [3], (iii) environmental assessment [4] and (iv) applications in forensic science [5]. The ‘gold standard’ in the analysis of low molecular weight volatile organic compounds in various matrices is gas chromatography (GC) coupled to mass spectrometric (MS) techniques. Liquid chromatography mass spectrometry (LC-MS) also serves for the analysis of small molecules, including non-volatile compounds, providing high throughput in complex mixtures. However, the separation of samples with complex chemical composition complicates the analysis of large number of samples to collect the molecular profiles. The optimization of separation conditions is a laborious and time-consuming procedure, even though the required amount of sample remains to be accounted in milliliters. In contrast, an untargeted approach in the application of MS techniques can be informative depending on the goal of the research, with low sample volume and providing fast data acquisition. In addition, mass spectrometry can serve as a versatile technique for the characterization of metal nanoclusters [6,7].

Laser desorption ionization/mass spectrometry (LDI-MS) was firstly utilized for the analysis of proteins with cobalt nanoparticles as an inorganic matrix by Tanaka et al. [8]. Noble metal nanoparticles are excellent candidates to assist ionization due to their UV-absorbing properties, chemical stability and reduced chemical background.

Many developments in the fabrication of various nanostructures based on noble metals such as gold and silver have been reported in the last decade [9]. One of the group of techniques employed for the fabrication of LDI nanoparticles-based substrates include wet and dry chemical methods. The wet chemical method is relatively simple and inexpensive; however, an uncontrolled aggregation of colloidal particles may occur, resulting in nonhomogeneous structures and thus significant signals inconsistency. Furthermore, the coffee ring effect can lead to a nonhomogeneous distribution of the analytes on the surface of nanostructures, resulting in poor reproducibility. To avoid aggregation, the preparation of the LDI substrates using wet chemical methods requires the application of stabilizers and reducing agents, which may complicate the spectra. On the other hand, dry methods such as electron beam lithography have provided well-controlled nanostructures and highly reproducible LDI performance [10]. However, these nanofabrication techniques require highly sophisticated and high-cost devices, as well as time-consuming and complicated procedures. Therefore, it is essential to prepare flexible, low-cost, time-saving and well-controlled nanostructured substrates for the analysis of low molecular weight profiles, with high sensitivity and reproducibility. Stainless steel is advantageous for the preparation of LDI-MS substrates due to its inexpensive price and relative chemical inertness. Moreover, H17 steel is available for commercial purchase in a wide variety of sheets of various thicknesses and sizes, which also simplifies its use as a substrate.

Chemical vapor deposition technique allows the synthesis of nanolayers of inorganic materials on the surface of 3D substrates [11]. The success of the deposition is dependent on the precursor utilized for synthesis; highly volatile, thermally stable compounds enable clean decomposition, potentially resulting in reduced chemical background when used in LDI-MS. Moreover, since the procedure is computer-controlled, it allows for the synthesis of well-controlled substrates and it is beneficial for target-to-target reproducibility [12]. Notably, well-controlled morphology may enhance the plasmonic properties of noble metals [13], potentially leading to the enhancement of LDI efficiency.

Despite the utilization of the CVD technique for fabrication of a nanostructured layer on various substrates, rare efforts have been focused on its application in LDI-MS. The CVD technique has been used previously for fabrication of carbon nanotubes [14] that were applied in the LDI-MS analysis of carbohydrates and amino acids. In addition, it has been utilized for synthesis of carbon nanowalls to be applied in analysis of fatty acids, lipids, saccharides, peptides [15], amino acids [16]; and graphene for analysis of carbohydrates [17]. To the best of our knowledge, metal nanostructures synthesized using the CVD technique have not yet been reported in LDI-MS analysis.

The aim of this study was to study the effect of the mass of the precursor on morphology and the LDI-MS efficiency of the obtained silver nanostructures towards low molecular weight analytes with utilization of chemical vapor deposition techniques. The LDI-MS plates synthesized with the proposed approach showed tunable sensitivity in both ion-positive and ion-negative modes. The plates can be used for the collection of molecular profiles and the analysis of small biomolecules with targeted approach at nanomolar and picomolar detection levels.

## 2. Materials and Methods

### 2.1. Reagents and Materials

Standards of low molecular weight compounds such as adonitol, glucose, fructose, shikimic acid, oleic acid, palmitic acid, cholesterol, methionine, serine, alanine and phenylalanine, all of the highest available purity, were purchased from Sigma Aldrich (Steinheim, Germany). Standards of various classes of lipids were purchased from Avanti Polar Lipids (Alabaster, AL, USA), including phosphatidylcholine 18:0 (<99%), lyso-PC, PE, PI and TG internal standard mixture (Ultimate SPLASH™, Avanti Polar Lipids, Alabaster, AL, USA). Solvents for the preparation of stock solutions of LC-MS grade quality (≥99.9%), such as water and chloroform, were purchased from Sigma Aldrich (Steinheim, Germany).

### 2.2. Synthesis and Characterization of LDI-MS Plates

For the synthesis of the LDI plates, stainless steel (H17) was cut to pieces 2.5 × 7.5 cm. The surfaces of the steel samples (substrates) were covered by the silver coating, consisting of densely packed silver nanoparticles and microparticles (AgPs). For this purpose, a chemical vapor deposition (CVD) technique was used under conditions described in Table 1. In our CVD experiments, Ag_5_(O_2_CC_2_F_5_)_5_(H_2_O)_3_ has been used as a precursor, the synthesis and physicochemical properties of which were earlier described [11,18,19,20,21]. The [Ag_5_(O_2_CC_2_F_5_)_5_(H_2_O)_3_] compound has also been used as a precursor in our CVD experiments; the synthesis and physicochemical properties of the compound were described earlier [11,21]. The fast and cheap synthesis of this precursor are among its advantages, as well as high structural stability of the silver(I) compound, allowing for long storage at room temperature without the access of light. The substrate surface preparation for the CVD process consisted of washing in an ultrasonic bath with distilled water containing a non-ionic surfactant for degreasing for 45 min (twice). Then, the substrate was immersed in the acetone (analytical grade) for 30 min, then distilled water for 10 min and, after drying in an Ar stream, it was placed in a CVD reactor. The morphology of created coatings was studied using a scanning electron microscope (SEM, Quanta 3D FEG, Houston, TX, USA). The structure of the AgPs films was investigated using an energy-dispersive X-ray diffractometer (Quantax 200 XFlash 4010) with a copper monochromator and CuKα radiation (λ = 0.15418 nm). XRD patterns were collected in the 2*Θ* range 10–80°, step 0.02° and time 20 sec. The Sartorius MCA2.7S-2S00-M microbalance (Sartorius Lab Instruments GmbH & Co. KG, Goettingen, Germany) has been applied to determine the weight of the reference sample before and after the CVD process. The stainless steel (H17) reference samples of sizes 1 × 1 cm were placed in the CVD reactor together with the investigated sample to obtain similar deposition conditions.

For the purposes of the MALDI experiments, Ag films were prepared in real time, and the storage time of samples (in a closed box, at room temperature and with limited access to light) was not longer than 2–3 days.

### 2.3. LDI-MS Analysis

The LDI-MS performance of the synthesized plates was evaluated by using stock solutions at concentration of 1 mg/mL and standard mixtures of various lipids. Stock solutions of adonitol, glucose, fructose, shikimic acid, methionine, serine, alanine and phenylalanine were prepared by dissolving a powder of each standard in water using 1.5 mL Eppendorf tubes. Stock solutions of cholesterol, oleic acid, palmitic acid and PC were prepared by dissolving a powder of each standard in chloroform using 1.5-mL amber glass vials and glass syringes for manual sample preparation (Agilent, Santa Clara, CA, USA). The standard mixtures of the various lipids were sonicated for 5 min prior to spotting to the target plate to avoid precipitation of lipids during storage. Subsequently, 1 µL of the stock solution of each compound and standard mixture was spotted to the synthesized LDI plates. LDI-MS analysis of low molecular weight compounds was carried out in both positive and negative ion-reflectron modes with the utilization of laser power at 80% in the mass range of *m*/*z* 60–1500. Analysis was performed using an UltrafleXtreme II MALDI-TOF-MS apparatus (Bruker Daltonics, Bremen, Germany) equipped with a modified neodymium-doped yttrium aluminium garnet (Nd:YAG) laser operating at 355 nm and frequency 2 kHz. The value of global attenuator offset was 30%, with a parameter set ‘5_ultra’ and the detector gain for reflector was 2.51× for all low molecular weight compounds except lipids. The following parameters were used for lipids: global attenuator offset 25%, with parameter set ‘4_large’ and the value of detector gain set to 30×. Mass calibration was performed using signals of silver using quadratic and cubic enhanced calibration methods individually for each spectrum. Reflector voltages accounted for 26.64 and 13.54 kV with first accelerating voltage set to 25.08 kV and the value for the second ion source voltage was 22.43 kV for the ion-positive mode. Reflector voltages for the ion-negative mode were 21.31 and 10.82 kV, with 20.07 kV as the first accelerating voltage and 17.97 kV as the second ion source voltage. Theoretical *m*/*z* values of the analyzed compounds were calculated by using ChemCalc program [22]. The number of laser shots was 2000 (4 × 500 shots) for each compound. LDI-MS targets were inserted into an MTP Slide-Adapter II (Bruker Daltonics, Bremen, Germany) and utilized for the collection of data. Heatmaps were prepared using GraphPad Prism software (version 8.0.1., San-Diego, CA, USA).

## 3. Results and Discussion

### 3.1. Characterization of LDI Plates

Our main idea was to study the dependency between the size of the deposited silver particles (AgPs), the coatings’ surface morphology and the LDI plates’ sensitivity to various low molecular weight compounds. For this purpose, the plates were subjected to characterization using scanning electron microscopy (SEM) and X-ray diffraction (XRD) techniques. The obtained results are presented in Table 2 and Figure 1 and Figure 2. The produced coatings consisted of densely packed metallic silver nano- and microparticles uniformly covering the entire surface of the LDI plates (Figure 1). The use of different masses of solid Ag precursor and similar deposition conditions (Table 1) enabled controlling the surface morphology of the deposited layers as well as controlling the size of the AgPs deposited. The analysis of SEM images revealed that the produced coatings could be divided into three groups from a morphological point of view (Figure 1). The use of the high Ag precursor concentrations (precursor weight: 35–100 mg) and accompanying coalescence effects led to a deposition of mainly micro-AgPs with irregular shapes. Films, which consisted of densely packed silver nanoparticles similar in shape to a sphere, were produced in the case of low precursor concentrations in vapors (precursor weight: 5–15 mg). Medium grain sizes of AgPs ranged from 50 to 240 nm for coatings produced using 5 and 15 mg of the precursor, respectively (Table 2, Figure 1). The further reduction in the applied precursor weight (up to 2.5 mg) caused a layer formation consisting of dispersed AgPs of dimeter ca. 150 nm. The registration of XRD patterns for studied samples confirmed the deposition of a pure form of metallic silver nanoparticles on the surface of steel substrates (Figure 2).

### 3.2. LDI-MS Performances of Silver Nanostructures for Low Molecular Weight Biomolecules

According to obtained results (Figure 3), all of the synthesized plates showed sensitivity to various low molecular weight compounds in both ion-positive and ion-negative modes. Low molecular weight analytes are biological molecules fulfilling various functions in the human organism and, thus, may serve as biomarkers of pathological processes, as has been suggested in numerous studies [1,23,24,25]. The studied compounds can be divided into three groups: water-soluble compounds, fatty acids and lipids, and amino acids. All compounds except lipids were ionized at nanomolar concentrations, while lipids showed sensitivity in both positive and negative modes at the picomolar level.

LDI-MS intensity has been shown to be different for various compounds depending on the mass of the precursor for both ion-positive and ion-negative modes. The differences in LDI-MS dependent on the applied mass of the precursor probably could be explained by the amount of deposited silver and the differences in the affinity of compounds towards silver nanostructures. Interactions of the analytes with nanostructured substrates may affect LDI-MS efficiency, and they can be characterized by complementary analytical techniques. A good example of such work was reported by Mandal et al. [26].

Figure 3 shows that the proposed approach allows for the synthesis of LDI plates with tunable sensitivity towards low molecular weight analytes. Water-soluble compounds such as adonitol, glucose and fructose showed higher intensity in the positive mode for all plates, as compared to shikimic acid, which demonstrated comparable intensities for plates AgPs0.02, AgPs0.11 and AgPs0.19 in the negative mode. Cholesterol was more efficiently ionized in the positive mode (Figure 3). Fatty acids showed LDI-MS intensity <2 × 10^5^ a.u. for all plates in both modes, except palmitic acid for plate AgPs0.06. In addition, the plates AgPs0.03, AgPs0.06 and AgPs0.11 in positive mode, in addition to AgPs0.06 and AgPs0.19, showed LDI-MS intensities close to 2 × 10 ^5^ a.u. for oleic acid. The plate AgPs0.06 was shown to be the most efficient for phenylalanine in both modes. Serine showed efficient ionization with intensity >2 × 10^5^ a.u for plates AgPs0.02 and AgPs0.19 in positive mode and AgPs0.02, AgPs0.06, AgPs0.19 and AgPs0.2 in negative mode. The ionization of methionine occurred with similar intensities <2 × 10^5^ a.u for all the plates in positive mode and intensities close to 2 × 10^5^ for the plates AgPs0.02, AgPs0.06 and AgPs0.11 in negative mode.

Notably, all of the compounds were detected in negative mode as ${\left[M\right]}^{-}\;$radical ions for all plates with high abundance. For some of the analytes, ${\left[M-H\right]}^{-}\;$species were also detected, as well as signals that may correspond to the fragments. Ionization in negative mode could occur via transfer of the hot electrons, as was suggested by Li et al., and hot electrons could be a source of charges in plasmonic metal nanostructures [27]. The occurrence of hot electrons can probably be explained by the well-controlled morphology of the silver nanostructures deposited by the CVD technique. To the best of our knowledge, hot electrons can be generated via localized surface plasmon resonance (LSPR) and interband transition [28]. Since LSPR depends strictly on the shape and size of the nanostructure, and all the plates showed sensitivity in ion-negative mode, the occurrence of hot electrons could also be related to interband transition. The occurrence of the signals corresponding to ${\mathrm{Ag}}_{n}^{-}$, which were observed for all the plates, also could be related to hot electrons [29].

Interestingly, the synthesized plates showed comparable efficiencies in both ion-positive and ion-negative modes, which might indicate an advantage for further applications of the plates. One possible interpretation is that the plates possess mechanisms of ion formation for both ion-positive and ion-negative modes. Substrate morphology allows for occurrence of both mechanisms: hot electrons transfer in negative mode and the cationization of silver in positive mode. The versatility of the obtained substrates for applications in ion-positive and ion-negative modes can be seen as potentially advantageous for the selective ionization of lipids in samples with rich composition. This may bring advantages for particular applications where selected classes of lipids are of interest; however, other classes of lipids with similar structures create interferences in mass spectra. Moreover, the selective isolation of lipids from samples with rich chemical compositions also represents an analytical challenge.

### 3.3. LDI-MS Performances of Silver Nanostructures for Lipids

The LDI-MS efficiency of the plates for lipids was evaluated using deuterated standard mixtures. Phosphatidylethanolamines (PEs) and triacylglycerols (TG) were detected in ion-negative mode at the picomolar level. Signals corresponding to ${\left[M-2H\right]}^{-}$ and ${\left[M-3H\right]}^{-}$ were assigned to molecular ions of PEs in negative mode (Figure 4). The plate AgPs0.02 provided the lowest intensities for PEs as compared to the plates AgPs0.11 and AgPs0.2, corresponding to 35 and 100 mg of precursor, respectively. In addition, plates AgPs0.02 and AgPs0.03 showed molecular ions as ${\left[M-3H\right]}^{-}$ and plate AgPs0.11 showed ${\left[M-2H\right]}^{-}$ for all PEs in the mixture. Plate AgPs0.2 provided ${\left[M-2H\right]}^{-}\;$for PEs with the highest values of monoisotopic mass (Figure 4; signals 4,5) and ${\left[M-3H\right]}^{-}$ for PEs assigned to 1, 2 and 3 (Figure 4). It could be suggested that such differences are related to the morphology and size of the obtained nanostructures. For example, ionization patterns of PEs were similar for the plates AgPs0.02 and AgPs0.03, both of which have specific features as compared to other plates, such as size (50 ± 10 nm) in the case of AgPs0.02, and morphology, i.e., isolated nanostructures, in the case of AgPs0.03. In contrast, plates AgPs0.11 and AgPs0.2, which showed higher intensities for PEs, consisted of microparticles with irregular shape. Moreover, affinities of lipids towards the nanostructured substrates also affect LDI-MS performance. It has been suggested that the high affinity of the analyte molecules towards the substrate can lead to reduced ionization efficiency, since it can decrease analyte desorption [30]. On the other hand, interactions between analyte molecules and substrate can promote selective and sensitive LDI ionization, as for example, was reported for olefins and silver nanoparticles [31]. Furthermore, the surface adsorption of the analytes may also play a role, thus suggesting that differences between the substrates could be also explained by differences in the surface area [32].

Triacylglycerols (TGs) were detected for the plate Ag0.11 in negative mode (Figure 5). All plates showed signals assigned to ${\left[M-2H\right]}^{-}$ for all triacylglycerols. Only the TG with mass 929.84 Da was not detected in negative mode.

Triacylglycerols were also detected for all plates in ion-positive mode (Figure 6). The heatmap (Figure 6) presents intensities for molecular ions assigned as an adduct such as ${\left[M+\mathrm{Ag}\right]}^{+}$/${\left[M+\mathrm{Na}\right]}^{+}$/${\left[M+K\right]}^{+}$, selected as those with the highest values of LDI-MS efficiency. The complete list of the values of *m*/*z* is presented in Appendix A. According to obtained results (Figure 6, Appendix A), plate AgPs0.02 provided signals corresponding to mostly ${\left[M+\mathrm{Ag}\right]}^{+}$ with the exception of TGs 18:1-17:1:18:1 and 18:1-19:2-18:1, which were detected as ${\left[M+H\right]}^{+}$ assigned to *m*/*z* 984.70 and 1010.71, respectively. The other plates showed signals corresponding to ${\left[M+\mathrm{Na}\right]}^{+}$ in most cases, and only selected plates showed signals corresponding to ${\left[M+\mathrm{Ag}\right]}^{+}$ and ${\left[M+H\right]}^{+}$ (Appendix A).

All of the lyso-PCs of the standard mixture were detected using plate AgPs0.04 (Figure 7). The signal at *m*/*z* 487.11 was assigned to ${\left[M+H\right]}^{+}$ and the signal at *m*/*z* 524.88 was assigned to ${\left[M+K\right]}^{+}$ corresponding to 15:0 lyso-PC. Other lyso-PCs such as 17:0 and 19:0 were detected as ${\left[M+H\right]}^{+}$ at *m*/*z* 515.02 and 543.02, respectively. The other plates, such as AgPs0.06 and AgPs0.11, provided molecular ions for only selected lyso-PCs. For example, the plate AgPs0.06 allowed to register 15:0 lyso-PC as ${\left[M+\mathrm{Na}\right]}^{+}$ at *m*/*z* 598.90 and ${\left[M+K\right]}^{+}$ at *m*/*z* 524.90, only where the value of global attenuator was 25% and parameter set ‘4_large’, while for plate AgPs0.04, registered molecular ions for all lyso-PCs and such conditions were not necessary. The plate AgPs0.11 provided molecular ions assigned to ${\left[M+K\right]}^{+}$ at *m*/*z* 524.83 corresponding to lyso-PC 15:0. The plate AgPs0.2 showed that the signal at *m*/*z* 525.35 was assigned to ${\left[M+K\right]}^{+}$ (15:0 *lyso*-PC), and the signal at *m*/*z* 538.66 probably corresponds to ${\left[M+\mathrm{Na}+H\right]}^{+}$ with relatively low intensity (S/N value equal to 6). ${\left[M+\mathrm{Na}+H\right]}^{+}$ species were probably less stable in the gas phase from the thermodynamic point of view.

The other classes of lipids (PC, PE and PI) might have undergone fragmentation, since their molecular ions were not detected. Due to the plasmonic properties of noble metal nanostructures, an excess of energy could lead to an enhanced fragmentation of lipids, which may complicate identification. A decrease in laser power did not result in obtaining signals corresponding to molecular ions. However, the plates could be used in the future for the collection of molecular profiles of samples, with an aim to reveal the differences and similarities between the samples, since fragments of various classes of lipids also can be characteristic.

## 4. Conclusions

The presented approach allows for the synthesis of LDI plates with tunable sensitivity for various classes of small biomolecules. The utilization of a chemical vapor deposition technique with various values of the mass of the precursor resulted in the formation of structures with sizes 50–330 nm and up to 1 μm with irregular shapes. Small biomolecules were detected at nanomolar concentrations, while lipids were detected at the picomolar level with a reduced chemical background. Sensitivity towards low molecular weight analytes in both ion-positive and ion-negative modes is an advantage for the applications of the plates for the collection of molecular profiles as well as targeted analysis.

## Figures and Tables

**Figure 1 materials-15-04660-f001:**
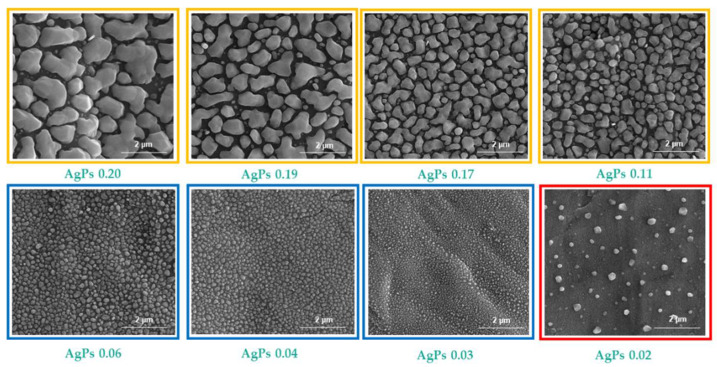
SEM images of AgPs films deposited on the surface of stainless steel (H17) substrates using CVD technique.

**Figure 2 materials-15-04660-f002:**
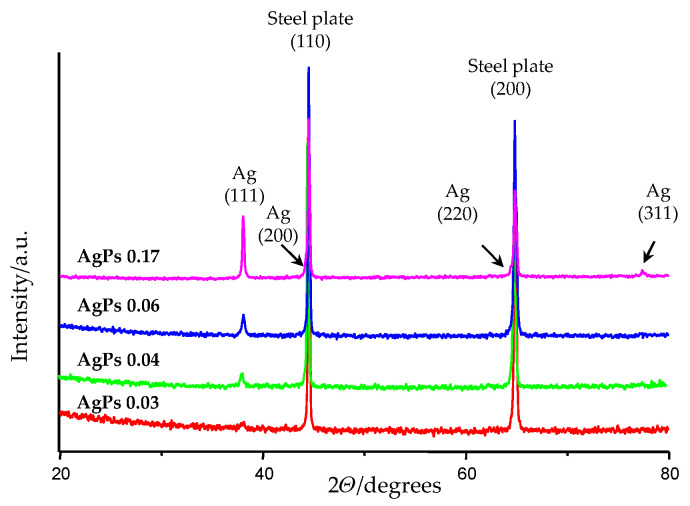
X-ray diffraction patterns of AgPs films deposited using CVD technique. A number of Bragg reflection peaks were observed at 2*θ* values of 38.2°, 44.3°, 64.3° and 77.6°, which are indexed to (111), (200), (220) and (311), respectively.

**Figure 3 materials-15-04660-f003:**
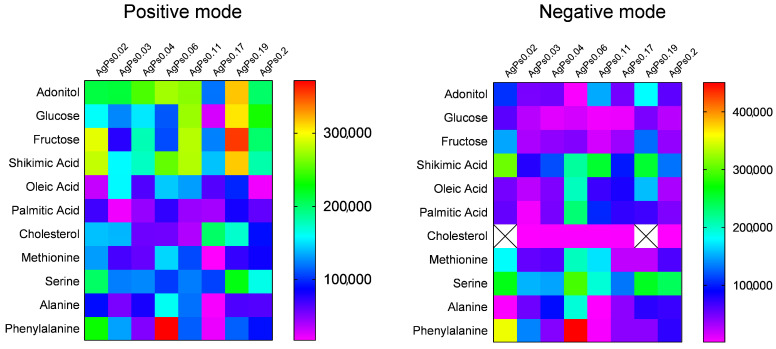
LDI-MS performances of the LDI-MS plates with various masses of silver for the analysis of low molecular weight compounds; molecular ions for all the compounds in positive mode were assigned as ${\left[M+{}^{107}\mathrm{Ag}\right]}^{+}$.

**Figure 4 materials-15-04660-f004:**
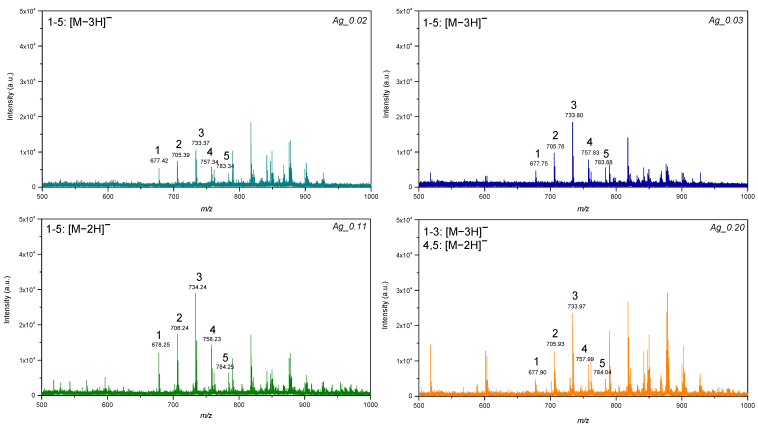
LDI-MS spectra of the standard mixture of phosphatidylethanolamines, where 1—17:0-14-1 PE (25 µg/mL); 2—17:0-16:1 PE (50 µg/mL); 3—17:0-18:1 PE (75 µg/mL); 4—17:0-20:3 PE (50 µg/mL); 5—17:0-22:4 PE (25 µg/mL), for the LDI plates AgPs0.02, AgPs0.03, AgPs0.11 and AgPs0.2.

**Figure 5 materials-15-04660-f005:**
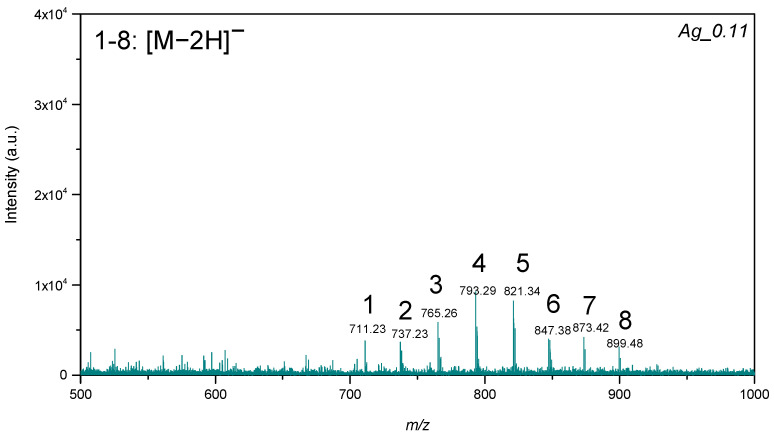
LDI-MS spectra of the standard mixture of triacylglycerols, where 1—14:0-13:0-14:0 TG (25 µg/mL); 2—14:0-15:1-14:0 TG (50 µg/mL); 3—14:0-17:1-14:0 TG (75 µg/mL); 4—16:0-15:1-16:0 TG (100 µg/mL); 5—16:0-17:1-16:0 TG (125 µg/mL); 6—16:0-19:2-16:0 (100 µg/mL); 7—18:1-17:1-18:1 TG (75 µg/mL); 8—18:1-19:2-18:1 TG (50 µg/mL).

**Figure 6 materials-15-04660-f006:**
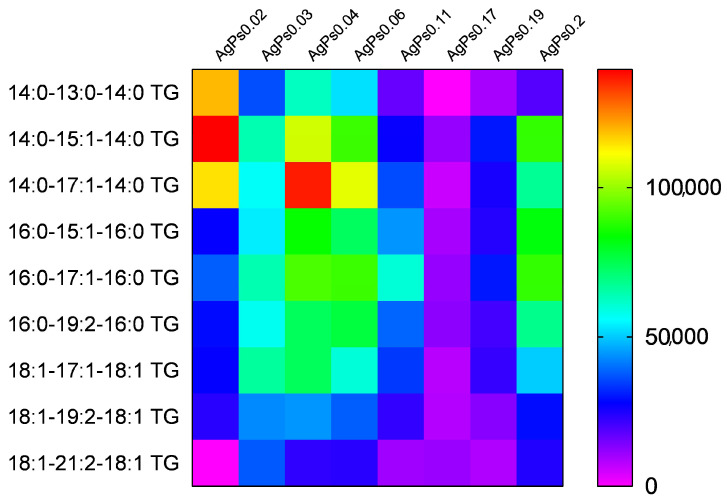
LDI-MS performance of LDI-MS plates with various masses of silver for analysis of standard mixture of triacylglycerols.

**Figure 7 materials-15-04660-f007:**
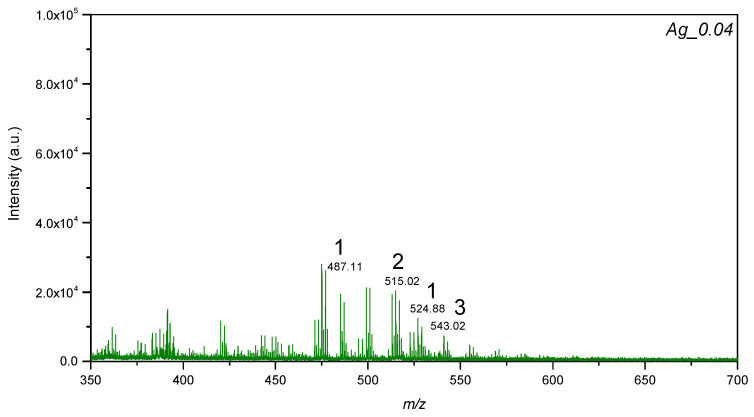
LDI-MS spectra of the standard mixture of lyso-phosphatidylcholines, where 1—15:0 LPC (25 µg/mL); 2—17:0 LPC (50 µg/mL); 3—19:0 LPC (25 µg/mL).

**Table 1 materials-15-04660-t001:** Deposition parameters of AgNPs’ coatings.

Precursor	Ag_5_(O_2_CC_2_F_5_)_5_(H_2_O)_3_
Precursor weight (mg)	2.5, 5, 10, 15, 35, 50, 70, 100
Vaporization temperature (T_V_) (°C)	230
Carrier gas	Ar
Total reactor pressure (p) (mbar)	3.0
Substrate temperature (T_D_) (°C)	290
Substrates	stainless steel (H17)
Deposition time (min)	60
Sample heating time (min)	30 (Ar/H_2_ (3:1%))

**Table 2 materials-15-04660-t002:** AgNPs films deposited by CVD technique.

Sample	Precursor Weight (mg)	Percentage Substrate Mass Increase after the CVD Process (wt.%)	AgPs Medium Grain Size (μm)
AgPs 0.20	100	0.20	0.7–2.8 ± 0.2–0.9
AgPs 0.19	75	0.19	0.5–1.7 ± 0.2–1.0
AgPs 0.17	50	0.17	0.2–0.7 ± 0.09–0.2
AgPs 0.11	35	0.11	0.33 ± 0.09
AgPs 0.06	15	0.06	0.24 ± 0.08
AgPs 0.04	10	0.04	0.15 ± 0.05
AgPs 0.03	5	0.03	0.05 ± 0.01
AgPs 0.02	2.5	ca. 0.02	0.15 ± 0.08

## Data Availability

All data generated or analyzed during this study are included in this published article (and its Appendix A).

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
