# Peer review of "Silver Nanostructured Substrates in LDI-MS of Low Molecular Weight Compounds"

_materials, 2022, doi:10.3390/ma15134660_

Round 1
Reviewer 1 Report
The authors have been developed an approach for preparation of silver nanostructured substrates applied LDI plates via CVD technique. The as-prepared LDI plates have tunable sensitivity for various low molecular weight compounds in both ion-positive and negative modes. Here are some questions for the authors before this article can be completely accepted in the Journal. The problems and suggestions are listed as follows.
(1) Why choose the stainless steel of H17 as the substrate? The advantages of H17 stainless steel could be provided in the section of introduction.
(2) Based on the P3, line 113 and 114: “The structure of AgPs films was investigated using energy-dispersive X-ray spectroscopy”. However, the detailed information was not provided in the paper. Please check the data.
(3) How about the stable of AgPs?
(4) There a lot of errors in the text. Such as the P5, line 180: SEM images; P7, line 235: performances.
(5) Additionally, please check the format of references. Such as P10, line 335: the reference [1] has no volume and page numbers. The reference of 3 has no page numbers. Please checked carefully.
Author Response
Response to Reviewers
Dear Editor,
Thank you for giving us the opportunity to submit the revised draft of the manuscript entitled ‘Silver nanostructured substrates in LDI-MS of low molecular weight compounds’ for publication in the journal Materials. We appreciate the time and effort that you and the reviewers dedicated for providing feedback for our manuscript and grateful for the insightful remarks that helped us to improve its quality. Based on these comments and thoughtful suggestions, we have made a careful revision of the original manuscript. A revised manuscript has been submitted, in which the modified sections are highlighted in red. Please see below, in blue, for a point-by-point response to the reviewers’ comments and concerns.
Reviewers’ comments to the Authors:
Reviewer 1
The authors have been developed an approach for preparation of silver nanostructured substrates applied LDI plates via CVD technique. The as-prepared LDI plates have tunable sensitivity for various low molecular weight compounds in both ion-positive and negative modes. Here are some questions for the authors before this article can be completely accepted in the Journal. The problems and suggestions are listed as follows.
(1) Why choose the stainless steel of H17 as the substrate? The advantages of H17 stainless steel could be provided in the section of introduction.
Response: Authors are grateful for this remark. The main advantages of application of stainless steel for preparation of LDI-MS substrates are cheap price and relative chemical inertness. Moreover, commercial purchase of H17 substrates also simplifies its use as a substrate. This information has been added to introduction following the Reviewer’s remark.
(2) Based on the P3, line 113 and 114: “The structure of AgPs films was investigated using energy-dispersive X-ray spectroscopy”. However, the detailed information was not provided in the paper. Please check the data.
Response: Authors are grateful for pointing this out. The sentence has been corrected in the revised version of the manuscript.
(3) How about the stable of AgPs?
Response: Indeed, stability of AgPs is important parameter for consideration since it may affect the LDI-MS performance and also determines the time of storage of the substrates. We have been studying silver nanoparticles obtained by the CVD method for a long time. After the deposition process, the samples were stored in closed boxes with limited access to the light. Earlier studies have shown that even after several months of storage under the above conditions, AgPs retained their structure, morphology and biological activity. For the purposes of the experiments described in this paper, Ag films were prepared in real time, and the time between CVD and MALDI experiments was not longer than 2-3 days. The relevant information has been added to experimental part. Moreover, attached XRD spectra (Fig. 2) showed only signals characteristic for silver and steel substrate.
(4) There a lot of errors in the text. Such as the P5, line 180: SEM images; P7, line 235: performances.
Response: The manuscript text has been grammar- and spell-checked following the Reviewer’s remark.
(5) Additionally, please check the format of references. Such as P10, line 335: the reference [1] has no volume and page numbers. The reference of 3 has no page numbers. Please checked carefully.
Response: The format of references has been checked and all the mistakes were corrected following the Reviewer’s comment.

Reviewer 2 Report
Sagandykova et al., performed an interesting study related to the synthesis of silver nanostructured substrates in LDI-MS of low molecular weight compounds. The manuscript is very well written and the information is properly presented. Thus, the results presented in the manuscript are novel and they should be published in Materials journal.
In order to improve the results reported I would suggest fulfilling the following comments:
1. The CVD technique would be a nice technique for the aims of the present research. The results obtained by using this technique might depend of several operation parameters e. g. temperature, temperature, time, etc. Is it possible to obtain the same substrate mass increasing by using different time and keeping the rest of conditions constant? In case a positive answer, why?
2. The results reported in table 2 revealed no trend of the medium grain size, what should explain this even by using the lowest precursor weight?
3. The conclusions of the present manuscript are supported mainly by the morphology (size) of the nanostructures. Taking into account the results reported in table 2 where no correlation of the grain size, these conclusions should be reformulated and should provide a stronger discussion to explain the differences between samples.
4. More characterization to see the Ag species should be very helpful for a better understanding of the different interactions with the molecules used.
With the comments above I consider that a minor revision of the manuscript must be done before acceptance for publication in the Materials Journal.
Author Response
Response to Reviewers
Dear Editor,
Thank you for giving us the opportunity to submit the revised draft of the manuscript entitled ‘Silver nanostructured substrates in LDI-MS of low molecular weight compounds’ for publication in the journal Materials. We appreciate the time and effort that you and the reviewers dedicated for providing feedback for our manuscript and grateful for the insightful remarks that helped us to improve its quality. Based on these comments and thoughtful suggestions, we have made a careful revision of the original manuscript. A revised manuscript has been submitted, in which the modified sections are highlighted in red. Please see below, in blue, for a point-by-point response to the reviewers’ comments and concerns.
Reviewers’ comments to the Authors:
Reviewer 2
Sagandykova et al., performed an interesting study related to the synthesis of silver nanostructured substrates in LDI-MS of low molecular weight compounds. The manuscript is very well written and the information is properly presented. Thus, the results presented in the manuscript are novel and they should be published in Materials journal.
In order to improve the results reported I would suggest fulfilling the following comments:
- The CVD technique would be a nice technique for the aims of the present research. The results obtained by using this technique might depend of several operation parameters e. g. temperature, temperature, time, etc. Is it possible to obtain the same substrate mass increasing by using different time and keeping the rest of conditions constant? In case a positive answer, why?
Response: We appreciate the given remark. We can guess that the Reviewer is concerned whether the deposition time, while maintaining the other conditions of the deposition process, can also provide diversified sizes of silver grains. Well, it is possible because the CVD process is not a self-limiting process - a longer deposition time will cause the silver grains to grow and, consequently, coalesce - until the formation of a practically continuous coating. For economic reasons - it is much better to optimize the process with the amount of precursor, than time - because these are, however, high-temperature and energy-consuming processes. We have a lot of experience in optimizing CVD processes - if a reviewer would be interested in this topic, please refer to the appropriate items, for example: (Radtke, A.; Grodzicka, M.; Ehlert, M.; Muzioł, T.M.; Szkodo, M.; Bartmański, M.; Piszczek, P Int. J. Mol. Sci. 2018, 19, 375, doi:10.3390/ijms19123962); (Piszczek P., Szłyk E., Chaberski M., Taeschner C., Leonhardt A., Bała W., Bartkiewicz K. Chem. Vap. Depos. 2005; 11:53–59, doi: 10.1002/cvde.200406323); (Szłyk E., Piszczek P., Chaberski M., Goliński A. Polyhedron. 2001; 20:2853–2861, doi: 10.1016/S0277-5387(01)00898-1); (Szłyk E., Piszczek P., Grodzicki A., Chaberski M., Goliński A., Szatkowski J., Błaszczyk T. Chem. Vap. Dep. 2001; 7:1–6). Relevant literature has been added to the text following the Reviewer’s remark.
- The results reported in table 2 revealed no trend of the medium grain size, what should explain this even by using the lowest precursor weight?
Response: We are sorry, but we have made a mistake in Table 2. For AgPs0.20, the grain size varies in the range of 0.7-2.8 mm (it was corrected in the revised version of the manuscript). In our opinion, the trends are clear, although they depend on the layer morphology. It is best visible for AgPs 0.06, 0.04, and 0.03 samples, where no coalescence processes occur during the coating formation and the surface is uniformly covered with spherical grains of a similar size. In the case of AgPs 0.02, the amount of the precursor in the gas phase is too small to form a uniform layer. Therefore, the nucleation of AgPs and the growth of individual grains were noticed. In the case of high concentrations of the precursor in the gas phase (AgPs 0.20, 0.19, 0.17, 0.11), the surface processes (adsorption, desorption, nucleation, diffusion, growth, coalescence) occur spontaneously, which leads to the formation of coatings consisting of various sizes grains and the irregular shapes. Extending the deposition time by 2-3 hours, we would obtain continuous layers of metallic silver, which was not the aim of the research.
- The conclusions of the present manuscript are supported mainly by the morphology (size) of the nanostructures. Taking into account the results reported in table 2 where no correlation of the grain size, these conclusions should be reformulated and should provide a stronger discussion to explain the differences between samples.
Response: Authors are thankful for the given comment. The discussion was constructed based on size as well as morphology of the nanostructured substrates since obtained substrates were divided into groups: (i) substrates with irregular structure, where it was challenging to determine average grain size and it could be explained by high mass of the precursor; (ii) substrates with more regular shape, where it was possible to compare the grain sizes, (iii) the substrate with the applied lowest mass of precursor (2.5 mg) that resulted in distinct morphology such as isolated nanoislands system. It was discussed in the section “3.1. Characterization of LDI plates”. In addition, it is necessary to study the effect of size and morphology of substrates further for particular applications that authors plan to accomplish in future. Finally, the discussion has been updated in the revised version of the manuscript.
- More characterization to see the Ag species should be very helpful for a better understanding of the different interactions with the molecules used.
Response: Authors appreciate the given Remark. Indeed, detailed characterization of the substrate may provide information on interactions of analytes with AgPs and its effect on the LDI-MS efficiency. First of all, we plan to apply LDI imaging technique to evaluate analytes distribution on the surface applying standard solutions of LMW compounds with various chemical structures. Furthermore, characterization of the substrate is challenging due to the need for in-situ analysis. For e.g., selected signals, characteristic for silver, were not observed for some of the plates due to signals coming from stainless steel. We also plan to apply Raman spectroscopy to assess the differences in signals (value of intensity and wavenumber) before (fresh substrate after synthesis) and after incubation with analytes.
With the comments above I consider that a minor revision of the manuscript must be done before acceptance for publication in the Materials Journal.
Response: Authors are thankful for dedication of time and effort of the Reviewer to assess our manuscript. Suggestions and comments given helped authors to improve the quality of manuscript and provided new insights into design of future experiments.

Reviewer 3 Report
It would improve the overall validity of your findings if you were to indicate the uncertainty of your data. This would allow the reader to truly see how different each of the measurements on the arrays were.
The use of error bars instead of colors is suggested to show that adjacent values are indeed statistically different. This is difficult to judge with the color map.
Also consider randomizing the locations of your analytes on the 2D array for future experiments.
Author Response
Response to Reviewers
Dear Editor,
Thank you for giving us the opportunity to submit the revised draft of the manuscript entitled ‘Silver nanostructured substrates in LDI-MS of low molecular weight compounds’ for publication in the journal Materials. We appreciate the time and effort that you and the reviewers dedicated for providing feedback for our manuscript and grateful for the insightful remarks that helped us to improve its quality. Based on these comments and thoughtful suggestions, we have made a careful revision of the original manuscript. A revised manuscript has been submitted, in which the modified sections are highlighted in red. Please see below, in blue, for a point-by-point response to the reviewers’ comments and concerns.
Reviewers’ comments to the Authors:
Reviewer 3
It would improve the overall validity of your findings if you were to indicate the uncertainty of your data. This would allow the reader to truly see how different each of the measurements on the arrays were.
Response: Authors are thankful for criticism and dedication of valuable time of the Reviewer to assess the manuscript. Indeed, uncertainty of data is important for reporting the findings. In case of LDI-MS, the spectra were collected manually from several shots throughout each spot and thus, value of intensity shows average spectra (500 shots × 4). In addition, shot-to-shot and target-to-target reproducibility will be tested in future for evaluation of possibility of applications of the plates for quantitative/semi-quantitative analysis. Authors appreciate the given comment and will consider it obligatory for design of future experiments.
The use of error bars instead of colors is suggested to show that adjacent values are indeed statistically different. This is difficult to judge with the color map.
Response: Authors are grateful for this comment. As it was mentioned above, the value of intensity reflects the average spectra since it was collected manually by shooting (500 shots × 4) throughout the whole spot. However, authors will consider also other methods for presentation of results in future, following the Reviewer’s remark.
Also consider randomizing the locations of your analytes on the 2D array for future experiments.
Response: Authors are grateful for this comment. Indeed, authors collected the spectra manually in this work and distribution of the analyte on the surface of AgPs may affect the LDI-MS performance. Therefore, we plan to study analytes distribution on the surface of AgPs using MS imaging techniques and standard solutions of various LMW analytes. Moreover, we would like to set the option of ‘random walk’ and automatic data acquisition to evaluate the effect of random shots on collection of final spectra and shot-to-shot reproducibility.

Reviewer 4 Report
Comments:
Manuscript (materials-1756760) entitled Silver nanostructured substrates in LDI-MS of low molecular weight compounds is scientifically well written and accepted for publication in materials after some minor issues.
- The author used the Ag5(O2CC2F5)5(H2O)3 as a precursor. Does the author use the previously described method for its synthesis.? Because Reference.9 is a review article. If possible, can add only appropriate references; who author exactly followed for its synthesis. Also, do the authors also measure its physicochemical properties.
- The precursor amount can directly affect the grain size. If yes, then different precursors can effect more effectively? Then why did the author choose Ag5(O2CC2F5)5(H2O)3 precursor?
- Figure 2 XRD of AgPs0.03, AgPs0.04, and AgPs 0.0.6 never showed the peak at (311) as can be seen in 0.17 . why the author never provided the samples XRD results, such as AgPs0.11 to AgPs 0.20, need to provide their data too. the author can take help from the latest article such as “Advanced Powder Technology Volume 32, Issue 9, September 2021, Pages 3388-3394”
- Also need to add XRD peak values with the plane in Figure-2, which will be more convenient for the readers.
Author Response
Response to Reviewers
Dear Editor,
Thank you for giving us the opportunity to submit the revised draft of the manuscript entitled ‘Silver nanostructured substrates in LDI-MS of low molecular weight compounds’ for publication in the journal Materials. We appreciate the time and effort that you and the reviewers dedicated for providing feedback for our manuscript and grateful for the insightful remarks that helped us to improve its quality. Based on these comments and thoughtful suggestions, we have made a careful revision of the original manuscript. A revised manuscript has been submitted, in which the modified sections are highlighted in red. Please see below, in blue, for a point-by-point response to the reviewers’ comments and concerns.
Reviewers’ comments to the Authors:
Reviewer 4
Manuscript (materials-1756760) entitled Silver nanostructured substrates in LDI-MS of low molecular weight compounds is scientifically well written and accepted for publication in materials after some minor issues.
The author used the Ag5(O2CC2F5)5(H2O)3 as a precursor. Does the author use the previously described method for its synthesis.? Because Reference.9 is a review article. If possible, can add only appropriate references; who author exactly followed for its synthesis. Also, do the authors also measure its physicochemical properties.
Response: Authors are grateful for the given comments. Details regarding the synthesis, structural characterization, and physicochemical properties of this compound are provided in Reference 16 and also our earlier papers concerning this group of Ag(I) compouns. We have added them to the references to be more credible. There are as follows: (Radtke, A.; Grodzicka, M.; Ehlert, M.; Muzioł, T.M.; Szkodo, M.; Bartmański, M.; Piszczek, P. Int. J. Mol. Sci. 2018, 19, 375, doi:10.3390/ijms19123962); (Piszczek P., Szłyk E., Chaberski M., Taeschner C., Leonhardt A., Bała W., Bartkiewicz K. Chem. Vap. Depos. 2005;11:53-59, doi: 10.1002/cvde.200406323); (Szłyk E., Piszczek P., Chaberski M., Goliński A. Polyhedron. 2001;20:2853–2861, doi: 10.1016/S0277-5387(01)00898-1); (Szłyk E., Piszczek P., Grodzicki A., Chaberski M., Goliński A., Szatkowski J., Błaszczyk T. Chem. Vap. Dep. 2001;7:1–6).
The precursor amount can directly affect the grain size. If yes, then different precursors can effect more effectively? Then why did the author choose Ag5(O2CC2F5)5(H2O)3 precursor?
Response: The presented research results are the beginning of a wider topic, which is the use of stainless-steel plates, with surface modified by AgNPs embedded by CVD for applications in LDI-MS. The choice of this precursor was determined by such factors as: (1) cheap and fast synthesis, (2) high stability of this compound, and thus easy storage, (3) no contamination with carbon or e.g. phosphorus of Ag layers, which we have often observed using Ag (I) complexes with tertiary phosphanes. Absence of contamination may allow to obtain the LDI-MS spectra with reduced chemical background that is advantageous for analysis in low mass range (< m/z 1000). In addition, continuing research on this issue, other silver precursors as well as deposition methods, such as ALD, will be tested.
Figure 2 XRD of AgPs0.03, AgPs0.04, and AgPs 0.0.6 never showed the peak at (311) as can be seen in 0.17 . why the author never provided the samples XRD results, such as AgPs0.11 to AgPs 0.20, need to provide their data too. the author can take help from the latest article such as “Advanced Powder Technology Volume 32, Issue 9, September 2021, Pages 3388-3394”
Response: Unfortunately, since the Ag grain diameter is reduced and the layer thickness is reduced, the intensity of the diffraction peaks decreases. This is true for all reflection peaks both high intensities, for e.g. Ag (111), and low intensity for Ag (311). The last peak is hidden in the instrumental noise. In future research, it will be necessary to use grazing incidence X-ray diffraction techniques in order to obtain more sensitive answer of apparatus.
Also need to add XRD peak values with the plane in Figure-2, which will be more convenient for the readers.
Response: Authors are thankful for pointing this out. In the revised version of the manuscript, 2Θ values have been added to the Figure 2 caption.

Reviewer 5 Report
The manuscript is focused on LDI-MS technique and the application of silver nanostructured substrated for studying the effect of the mass of precursor on morphology and LDI-MS efficiency with utilization of chemical vapor deposition technique.
The study represents an interesting area of mass spectrometric techniques but outlines practical implications in many areas of research and industry.
The paper is well written, in literar English. It is well structured, giving the reader clear idea of the main goal, the methodology and the results.
The figures are of high quality, informative and well defined.
I have the following minor remarks:
1. The sentence in lines 35-37 does not sound clear, it should be edited so that the reader is able to anderstand which is the targeted item - the compound or the disease.
2. XRD analysis should be included in the Materials and methods chapter to correspond to the Results section.
Author Response
Response to Reviewers
Dear Editor,
Thank you for giving us the opportunity to submit the revised draft of the manuscript entitled ‘Silver nanostructured substrates in LDI-MS of low molecular weight compounds’ for publication in the journal Materials. We appreciate the time and effort that you and the reviewers dedicated for providing feedback for our manuscript and grateful for the insightful remarks that helped us to improve its quality. Based on these comments and thoughtful suggestions, we have made a careful revision of the original manuscript. A revised manuscript has been submitted, in which the modified sections are highlighted in red. Please see below, in blue, for a point-by-point response to the reviewers’ comments and concerns.
Reviewers’ comments to the Authors:
Reviewer 5
The manuscript is focused on LDI-MS technique and the application of silver nanostructured substrated for studying the effect of the mass of precursor on morphology and LDI-MS efficiency with utilization of chemical vapor deposition technique.
The study represents an interesting area of mass spectrometric techniques but outlines practical implications in many areas of research and industry.
The paper is well written, in literar English. It is well structured, giving the reader clear idea of the main goal, the methodology and the results.
The figures are of high quality, informative and well defined.
I have the following minor remarks:
- The sentence in lines 35-37 does not sound clear, it should be edited so that the reader is able to anderstand which is the targeted item - the compound or the disease.
Response: We thank the Reviewer for careful reading of the manuscript and constructive remarks. All the comments were taken on board to improve and clarify the manuscript. The sentence in lines 35-37 has been edited to improve its clarity for readers.
- XRD analysis should be included in the Materials and methods chapter to correspond to the Results section.
Response: Details of XRD analysis have been added to the section Materials and Methods. Authors are grateful for pointing this out.

Reviewer 6 Report
In this manuscript, Gulyaim Sagandykova et al reported the LDI-MS of low molecular weight of small biomolecules of various classes (fatty acids, amino acids, water-soluble metabolites) using Silver nanostructured substrates. The work is interesting and well organized. I recommend the publication of the work in journal of materials after some changes.
1. The materials were prepared by immersion in acetone (analytical grade) for 30 min, the acetone in ppm is hard to remove from the GC system, which will lead to some impurity.
2. As shown in Figure 1, the particle size in AgPs0.02 is larger than AgPs0.03, why?
3. Some recent related literatures on MS should be cited, e.g. 10.1007/s11426-022-1267-5, etc, and MS on Ag species, e.g. 10.1007/s12274-021-3928-4.
4. In Figure 3, what is the different of the analysis in positive and negative modes. And please discuss it in more detailed.
Author Response
Response to Reviewers
Dear Editor,
Thank you for giving us the opportunity to submit the revised draft of the manuscript entitled ‘Silver nanostructured substrates in LDI-MS of low molecular weight compounds’ for publication in the journal Materials. We appreciate the time and effort that you and the reviewers dedicated for providing feedback for our manuscript and grateful for the insightful remarks that helped us to improve its quality. Based on these comments and thoughtful suggestions, we have made a careful revision of the original manuscript. A revised manuscript has been submitted, in which the modified sections are highlighted in red. Please see below, in blue, for a point-by-point response to the reviewers’ comments and concerns.
Reviewers’ comments to the Authors:
Reviewer 6
In this manuscript, Gulyaim Sagandykova et al reported the LDI-MS of low molecular weight of small biomolecules of various classes (fatty acids, amino acids, water-soluble metabolites) using Silver nanostructured substrates. The work is interesting and well organized. I recommend the publication of the work in journal of materials after some changes.
Response: Thank you so much for appreciation of our manuscript, time and effort dedicated to its review. Authors also are grateful for useful and constructive comments that helped authors to improve the manuscript.
- The materials were prepared by immersion in acetone (analytical grade) for 30 min, the acetone in ppm is hard to remove from the GC system, which will lead to some impurity.
Response: Authors are grateful for the given comment. The synthesis temperature ensures evaporation of solvents used for pre-cleaning of substrates. In addition, LDI-MS spectra demonstrated reduced chemical background in low mass region (< m/z 1000).
- As shown in Figure 1, the particle size in AgPs0.02 is larger than AgPs0.03, why?
Response: Authors are grateful for this comment. Indeed, the particle size in case of AgPs0.02 is larger than AgPs0.03 since notations of the plates were generated using the mass of silver, not the particle size. Application of the particle size is more reasonable as it was noticed by the Reviewer, however we believe that mass of silver can simplify reading of the text since notations are unified, and have shorter length. Moreover, we made a mistake in Table 2. For AgPs0.20, the grain size varies in the range of 0.7-2.8 mm (it was corrected in the revised version of the manuscript).
- Some recent related literatures on MS should be cited, e.g. 10.1007/s11426-022-1267-5, etc, and MS on Ag species, e.g. 10.1007/s12274-021-3928-4.
Response: Authors thank the Reviewer for pointing out interesting literature regarding MS applications. The literature has been added to the text following the recommendation.
- In Figure 3, what is the different of the analysis in positive and negative modes. And please discuss it in more detailed.
Response: Authors appreciate the comment given by the Reviewer and thankful for taking time to assess the manuscript. First of all, analytes are ionized via different mechanisms in positive and negative modes. To the best of our knowledge, generation of hot electrons may contribute to ion generation in negative mode, while ion formation occurs via cationization with silver in positive mode. Discussion has been updated in the revised version of manuscript following the Reviewer recommendation.

Round 2
Reviewer 1 Report
My concerns have been addressed carefully. It can be publishable in this form.